# Differentiating Human Pluripotent Stem Cells to Cardiomyocytes Using Purified Extracellular Matrix Proteins

**DOI:** 10.3390/bioengineering9120720

**Published:** 2022-11-22

**Authors:** Ashlynn M. Barnes, Tessa B. Holmstoen, Andrew J. Bonham, Teisha J. Rowland

**Affiliations:** 1Department of Molecular, Cellular, and Developmental Biology, University of Colorado Boulder, Boulder, CO 80309, USA; 2Department of Chemistry & Biochemistry, Metropolitan State University of Denver, Denver, CO 80217, USA

**Keywords:** developmental biology, induced pluripotent stem cells, human embryonic stem cells, cardiomyocytes, heart models, extracellular matrix, differentiation, cardiac tissue engineering, iPSC, hESC

## Abstract

Human embryonic stem cells (hESCs) and induced pluripotent stem cells (iPSCs) can be differentiated into cardiomyocytes (hESC-CMs and iPSC-CMs, respectively), which hold great promise for cardiac regenerative medicine and disease modeling efforts. However, the most widely employed differentiation protocols require undefined substrates that are derived from xenogeneic (animal) products, contaminating resultant hESC- and iPSC-CM cultures with xenogeneic proteins and limiting their clinical applicability. Additionally, typical hESC- and iPSC-CM protocols produce CMs that are significantly contaminated by non-CMs and that are immature, requiring lengthy maturation procedures. In this review, we will summarize recent studies that have investigated the ability of purified extracellular matrix (ECM) proteins to support hESC- and iPSC-CM differentiation, with a focus on commercially available ECM proteins and coatings to make such protocols widely available to researchers. The most promising of the substrates reviewed here include laminin-521 with laminin-221 together or Synthemax (a synthetic vitronectin-based peptide coating), which both resulted in highly pure CM cultures. Future efforts are needed to determine whether combinations of specific purified ECM proteins or derived peptides could further improve CM maturation and culture times, and significantly improve hESC- and iPSC-CM differentiation protocols.

## 1. Introduction

Using human embryonic stem cells (hESCs) [1] or human induced pluripotent stem cells (iPSCs) [2,3] as a readily available, unlimited source for creating cardiomyocytes and related tissues holds immense potential to address many unmet needs in the cardiology field. The heart, due to its internal location and essential function, is challenging to study; hence, creating accurate cellular models would be ideal for investigating heart development and disease, including high-throughput drug screening efforts and the generation of patient-specific models using patient iPSC-derived cardiomyocytes (iPSC-CMs). Additionally, while many people require a heart transplant, donors are limited; regenerative medicine-based solutions to this using hESC-derived cardiomyocytes (hESC-CMs) or iPSC-CMs to generate replacement cardiac tissues or “patches” are extremely appealing. However, while human hESC- and iPSC-CMs are promising tools for meeting both unmet needs, existing protocols for differentiating iPSCs or hESCs into CMs require xenogeneic (xeno) (animal-derived) and undefined culture components. Additionally, these protocols are relatively inefficient and not highly reproducible; they result in cultures that can vary from 20–85% CMs, which are relatively immature and significantly “contaminated” with other, non-CM cell types [4,5,6,7]. A maturation process can improve the maturity of the CMs but requires months of continuous culture to do so [8,9,10]. While these protocols most commonly use a mouse tumor-derived substrate, Matrigel, recent studies by multiple groups have suggested that the differentiation process may be accomplished using specific, purified extracellular matrix (ECM) proteins in place of Matrigel, offering defined, xeno-free substrate alternatives [4,5,6,11].

This review summarizes the findings of recent studies on the impact of ECM proteins on hESC- and iPSC-CM differentiation, with the aim of giving recommendations for defined and xenogeneic-free substrate options that could be easily and routinely adopted by those utilizing hESC- and iPSC-CMs in their research. The most widely cited hESC- or iPSC-CM differentiation protocol employs culture on Matrigel [12], which is a “feeder-free” substrate. However, Matrigel is a basement membrane preparation of the Engelbreth-Holm-Swarm mouse sarcoma tumor, grown in and extracted directly from mice to produce Matrigel, and contains a mixture of mouse proteins (including ECM proteins) and growth factors that varies from lot to lot [13,14,15]. This makes Matrigel nonideal for regenerative medicine efforts, such as cardiac tissue transplants, where these foreign proteins may stimulate a host immune response and immune rejection. Additionally, analyses of hESC- and iPSC-CMs differentiated on Matrigel may be contaminated with unwanted mouse proteins, complicating interpretation of results, sometimes invalidating them. Historically, undifferentiated hESCs or iPSCs were maintained on mouse or human feeder fibroblasts, though most researchers have transitioned to feeder-free systems, with Matrigel being the most common of these [15,16]. Matrigel was likely chosen to be used for initial CM differentiation efforts because of its common use in undifferentiated feeder-free culture systems. The ECM is known to play a role in supporting hESC and iPSC differentiation in general, including differentiation to specific target cell types [17,18,19,20,21,22,23,24], though its role in CM differentiation specifically is poorly understood. 

Different approaches for altering the substrate during hESC- and iPSC-CM differentiation have been investigated—including optimizing 3D, hydrogel-based systems and synthetic scaffolds as well as using decellularized human and animal tissues for potential cardiac tissue transplant efforts—though the focus of this review will be on the use of individual, purified, ECM proteins as a coating of culture vessels or commercially available substrates to make the findings summarized here easily adoptable by most researchers. While natural hydrogels (often containing ECM proteins) and synthetic hydrogel systems (often containing ECM-based peptides, such as Arginine-Glycine-Aspartic acid (RGD)) have been tested for their ability to support CM differentiation and engraftment, hydrogel studies are beyond the scope of this review; instead, the authors recommend readers refer to several recent reviews on this topic [25,26,27,28,29,30,31,32,33,34,35]. The primary ECM proteins covered here include laminins, collagens, fibronectin, and vitronectin, all of which are expressed by human cardiac tissues along with integrin receptor counterparts. The most promising hESC- and iPSC-CM differentiation results have been reported with protocols using human recombinant laminin-521 and laminin-221 protein coatings combined or Synthemax, a commercially available, synthetic human vitronectin peptide-based coating system; these protocols generated CM cultures ≈80–90% [11] or 97% [5] pure, respectively. These substrates may provide suitable alternatives to Matrigel for a wide range of researchers seeking to create a more defined, xeno-free, and biomimetic hESC- or iPSC-CM differentiation system. Future efforts may build upon these findings to better determine whether such substrates and ECM proteins in combination (and potentially at different concentrations or ratios) may produce a more supportive differentiation environment, potentially further improving purity, yields, and maturation of resultant hESC- and iPSC-CMs. This may aid in overcoming these challenges in the field, and lead to improved cellular cardiac models for development and disease studies (including high-throughput drug screening efforts), as well as enhanced 3D synthetic bioengineered scaffold systems for therapeutic cardiac tissue transplants.

## 2. hESC- and iPSC-CM Differentiation Protocols and Limitations

### 2.1. Overview of Protocols

The most highly cited hESC- and iPSC-CM differentiation protocol, Lian et al., 2013, is an adherent, 2D, plate-based protocol that involves seeding undifferentiated hESCs or iPSCs onto a Matrigel-coated plate [12]. In this protocol, CM differentiation is promoted by temporally modulating Wnt/β-catenin signaling through media supplements [12]. For protocol details, it is recommended that the reader refer directly to Lian et al., 2013, which is a published Nature Protocol. A second, alternative hESC- and iPSC-CM differentiation approach has been developed that starts by forming non-adherent embryoid bodies (EBs), though this method typically results in lower purity CM cultures, as discussed in more detail in Section 2.2 Protocol Limitations. Finally, a third and more promising approach developed, termed the “Matrigel Sandwich,” involves culturing hESCs or iPSCs sandwiched between two layers of Matrigel [4,36]. The Matrigel Sandwich method appears to yield higher purity CM cultures, though it still employs Matrigel. For protocol details, it is recommended that the reader refer directly to Zhang et al., 2022, which includes schematic figures. While the limitations of all three approaches are explored in the following section, the remainder of this review will focus on 2D adherent and Matrigel Sandwich-type methods, as these best investigate the impact of the ECM on hESC- and iPSC-CM differentiation.

### 2.2. Protocol Limitations

Current hESC- and iPSC-CM differentiation protocols are primarily limited because (1) the differentiation culture times are lengthy, (2) the resultant hESC- and iPSC-CMs they produce are relatively immature, (3) the resultant CM cultures are contaminated by the presence of non-CM cell types, and (4), as described in Section 2.1 above, all 2D adherent and Matrigel Sandwich-type protocols employ Matrigel. While in Lian et al., 2013, cellular contractions may appear by day 12 of the differentiation process, and additional hallmarks of CMs should be present at day 14 (including expression of several key CM markers) [12], weeks of additional differentiation and culture are typically required to improve purity and maturity of these CM cultures; this is discussed in detail in the following paragraph. However, it is worth noting that some researchers and their experimental goals may not require mature CMs, and more immature, fetal-like CMs may suffice; the authors encourage researchers to decide what degree of maturity is necessary for their experimental goals. 

In Yang et al., 2014, characteristics of adult CMs and immature CMs derived from hESCs and iPSCs were thoroughly and concisely summarized; we invite the readers here to refer to this previous review for details on, including metrics of, key maturation differences between hESC- and iPSC-CMs and adult CMs, as many of these specifics are beyond the scope of this review [8]. Another recent review, Sun and Nunes, 2017, also provided detailed comparisons of hESC- and iPSC-CMs to adult CMs in terms of maturation [37]. The comparison in Yang et al. included differences in CM size, shape, sarcomere structures, electrophysiological properties, binucleation, and abundance and distribution of mitochondria, as well as several cardiac genes that are upregulated in the adult heart compared to hESC- and iPSC-CMs, related to sarcomere and ion transporter functions [8]. One potential solution to make hESC- and iPSC-CMs more mature is to lengthen the time in which they are cultured. While researchers may often culture hESC- or iPSC-CMs for 60 to 90 days before assessing CM-related features, an interesting study by Kamakura et al. reported that even after 360 days in culture, the resultant CMs are still not as mature as adult heart CMs [9]. Specifically, after 180 days in culture, CM cultures gained more tightly packed, mature myofibrils with apparently mature Z-, A-, H-, and I-bands, but no M-bands present; M-bands were detected only after 360 days of culture, though M-band related gene expression levels were lower than those found in normal adult heart samples [9]. Agreeing with the need for lengthy culture times to improve maturity of these cells, Dias et al., 2018, similarly differentiated iPSCs into CMs (using the Lian et al., 2013 protocol) and found that resultant cardiac fibers (measured via immunocytochemistry staining of cardiac troponin T (cTnT) bands) showed peak cardiac fiber density around day 90 (66%, increasing from 52% at day 15), showing an improvement of sarcomere definition over this time, and the percentage of cells expressing cTnT decreased after the day 90 timepoint (quantified via flow cytometry), resulting in a percentage of cardiomyocytes at day 120 similar to what is found in mature, adult human heart tissue (22% reported) [10]. Overall, considerable culture time lengths appear to be required to fully mature hESC- and iPSC-CMs to function similar to adult human heart tissue. Reducing lengthy culture times is ideal for researchers and GMP manufacturing purposes; not only would this decrease the cost of the research and potential downstream manufacturing (both in terms of labor and reagents consumed), but risks associated with microbial contamination (the cells are typically cultured without antibiotics and antimycotics) would be decreased as well. However, recently successes have been reported in improving CM maturation by altering the differentiation culture conditions to recapitulate key developmental aspects; this is a promising approach that may overcome lengthy culture times [38].

Regarding contamination of harvested hESC- and iPSC-CM cultures with non-CM cells, most current protocols are considered successful if they achieve >80% purity when cells are harvested, though this means that a significant portion of the harvested cells are not CMs. As with determining how mature one’s CMs need to be, the authors here similarly encourage researchers to decide what percentage of contaminating non-CMs is acceptable for reaching a given experimental goal, keeping in mind that the human heart itself is also a mix of CM and non-CM cells. Lian et al. reported relatively high efficiencies, with 80–98% of cells being positive for cTnT after 2 weeks of differentiation [12]. Efficiencies of other 2D, adherent plate-based studies are discussed later, in Section 4, in terms of how the ECM coating impacts these efficiencies. 3D methods using EB formation to generate hESC- and iPSC-CMs, including in combination with polymeric hydrogels, have typically been less efficient than 2D, adherent plate-based protocols; CM purities reported are usually <30% [5,39,40,41,42], with some more efficient protocols reaching >60% CMs [43,44]. EBs are formed by aggregating pluripotent hESCs or iPSCs, typically seeding hundreds to thousands of cells per well on a 96-well plate or per microwell on a microwell plate. For a recent review on such 3D hESC- and iPSC-CM differentiation approaches, including generating cardiac microtissues and comparison of different cardiac cell types produced, see [45]. In addition to the 2D monolayer and 3D EB-based approaches, a third method for generating hESC- and iPSC-CMs involving culturing a monolayer of PSCs on Matrigel and then overlaying that monolayer with additional Matrigel, or the so-called “Matrigel Sandwich” method [36], has reported very high purities of CMs, up to 98% (based on cTnT+ expression after 30 days of differentiation). While it results in highly pure CMs, a potential limitation to the Matrigel Sandwich method is it may be more technically challenging than a simple plate-coating approach. Zhang and colleagues later performed additional similar overlay differentiation experiments with different ECM proteins [4], which are discussed in detail in Section 3.2.

As all 2D adherent and Matrigel Sandwich-type protocols for hESC- and iPSC-CM differentiation utilize Matrigel, it is important to discuss the limitations of this substrate. Matrigel is the most established and common feeder-free system for undifferentiated hESC and iPSC culture [15,16], and used in the most highly cited hESC- and iPSC-CM differentiation protocol and several derivatives [12]. Matrigel itself, extracted from the Engelbreth-Holm-Swarm mouse tumor grown in mice, contains a mixture of ECM proteins and growth factors, including laminin-111, collagen IV, heparin sulfate proteoglycan, nidogen-1 (formerly known as entactin), and various active growth factors that additionally impact cellular migration and organization [13,46]. Ultimately, while it is a generally supportive substrate, Matrigel does not accurately reconstruct any in vivo native ECM environment [21]. Furthermore, because it is comprised of mouse components, Matrigel is not xeno-free, and because its components vary in concentration from lot to lot, it is not a “defined” product either. (Being a defined product indicates that the concentrations of all components are known and do not vary). Overall, many researchers are undertaking efforts to utilize alternative substrate systems [23]. Of note, harvested CMs have been shown to mature better on Matrigel compared to other individual ECM proteins, suggesting that an ideal, xeno-free, post-differentiation maturation process would likely require culturing harvested CMs on a combination of different ECM proteins together [47]. 

## 3. Using the Extracellular Matrix to Improve hESC- and iPSC-CM Differentiation

### 3.1. Overview: Expression of ECM Proteins and Integrins in the Human Heart

The two main macromolecule classes that comprise the ECM are proteoglycans and fibrous proteins [21], of which this review will focus on fibrous proteins as they interact with cells through integrins, mediating downstream intracellular signaling pathways. Proteoglycans, which are heavily glycosylated proteins, form a hydrated gel within the ECM space. Fibrous proteins provide structural elements, including tensile strength and support repeated stretching, cell migration, and chemotaxis, and additionally regulate cell adhesion and cellular and tissue changes that occur during normal development [21,36]. Consequently, the ECM composition changes depending upon developmental stage and tissue type, as well as disease state and progression. All cell types interact with these fibrous ECM proteins through integrins, which mediate downstream signaling pathways required for cell adhesion, proliferation, apoptosis, and differentiation [17,48,49,50]. Integrins are multifunctional transmembrane heterodimer receptors, each comprised of an α- and a β-subunit, where the specific αβ heterodimer combination dictates which ECM protein(s) the integrin receptor interacts with extracellularly [48,49,50]. The undifferentiated state of hESCs and iPSCs itself depends upon expression and interaction of key integrins with their specific ECM counterpart proteins [13,15,51,52,53,54,55]. While the ECM is known to play roles in supporting hESC and iPSC differentiation [17,18,19,20,21,22,23,24], it is poorly understood how it supports CM differentiation specifically. To create a xeno-free, defined hESC- and iPSC-CM differentiation process, several groups have investigated the impact of defined, purified ECM proteins on differentiation, with a commonly applied strategy being to attempt to mimic the native human cardiac microenvironment, based on ECM and integrin expression profiles both during development and in the adult heart [4,5,6,11]. Reported expressions of specific integrin subunits and their ECM protein binding partners in human cardiac tissue is summarized in Table 1 [11,54,56,57,58]. However, because the CM differentiation process is a multi-stage, dynamic, developmental process—with cells progressing from pluripotent, to mesodermal, and then cardiac (immature to mature) states—the supportive ECM microenvironment undoubtedly changes throughout differentiation. As described in Section 3.2 below, ECM proteins that effectively maintain pluripotent hESCs and iPSCs, such as vitronectin and laminin-521, may be unsupportive of CM differentiation, and similarly ECM proteins that support CM cultures may be unsupportive of pluripotent hESCs and iPSCs and therefore fail to support CM differentiation initiation. Ultimately, a dynamic ECM microenvironment that is progressively altered (both in ECM protein composition and concentrations) to support the different cellular stages as they transition from pluripotency to a mature CM fate would be ideal, complementing the ECM proteins generated and secreted by these cells themselves. Functional studies to date have focused on the ECM proteins that the pluripotent cells are seeded on, at the beginning of CM differentiation. These functional studies have investigated different ECM proteins or substrates for their ability to support and impact hESC- and iPSC-CM differentiation, primarily using laminins, collagens, vitronectin, and fibronectin. The focus of the remainder of this review will be on summarizing the findings of these expression and functional studies, and the different ECM proteins investigated therein.

### 3.2. Laminins

Laminins are ECM proteins that serve many important cellular functions, including mediating cell adhesion, migration, angiogenesis, and differentiation [59]. Each laminin is a trimeric protein, comprised of three subunits, an α, β, and γ chain, that make up at least 16 different possible isoforms in mammals [11], with the exact subunit combination determining which integrin(s) binds that laminin ligand. In human cardiac tissue, laminin expression is constant throughout life—it is highly expressed during fetal cardiac development and in adult cardiac tissue [63]—with expressed laminins including laminin-211, -221, -411, -421, -511, and -521 (Table 1). All of these laminins are ligands for the α3β1 and α6β1 integrin heterodimers, which are also expressed in adult cardiac tissue. Some of these laminins are also ligands for the α7β1 integrin heterodimer (specifically laminin-211 and -221), which is additionally expressed in human cardiac tissue (Table 1). In particular, the laminin subunit *LAMB2* (also called β2) was found to be the most highly expressed laminin subunit investigated in 108 non-diseased human donors, with the next most highly expressed laminin subunits found to be *LAMA2*, *LAMB1*, and *LAMC1*, which encode the α2, β1, and γ1 laminin subunits, respectively [11]. Together, this study suggests that laminin-221 (comprised of the α2, β2, and γ1 laminin subunits) is the most highly expressed laminin in non-diseased human hearts, followed by laminin-211. Altogether, laminin-211, -221, -411, -421, -511, and -521 may be involved in normal cardiac-ECM interactions, based on expression levels of the laminins and integrin receptors. 

While based on expression levels, laminin-211, -221, -411, -421, -511, and -521 may be involved in normal cardiac-ECM interactions (Table 1), only a subset of these (and additional) laminins have been used in multiple investigations testing their ability to support and effect hESC- and iPSC-CM differentiation. We were unable to find publications using laminin-411 or -421 coatings, though a recent study reported laminin-411 to be important for increasing iPSC-derived endothelial differentiation via Wnt signaling [64] and laminin-111 and -421 have been used as part of a laminin-entactin complex to generate hESC-derived heart organoids, suggesting laminin-411 and -421 merit additional coating-based investigations [65]. Studies focused on differentiating hESC- and iPSC-CMs on laminin-111, -211, -221, -511, and -521 will be explored in the remainder of this section. Of note, many of the laminins used in these studies were human recombinant proteins purchased commercially from Biolamina or other commercial sources.

While laminin-111 is likely only lowly expressed in adult human cardiac tissue, because it is the most abundant component of Matrigel, its purified form independent of Matrigel has been investigated for its ability to support hESC- and iPSC-CM differentiation. The manufacturer of Matrigel, Corning, states that Matrigel is comprised of specifically 60% laminin-111 and 30% collagen IV. Laminin-111 interacts with integrins α1β1, α2β1, α6β1, α6β4, α7β1, and αVβ8, while collagen IV interacts with integrins α3β1 and αVβ8 [59,60,61,62]. Because Matrigel is a mixture of different ECM proteins, it is challenging to determine which proteins—and integrin interactions—are most important for supporting and promoting CM differentiation. We previously found purified mouse laminin-111 to support expansion of undifferentiated human pluripotent stem cells, as others have also reported [4,17,53]. Zhang and colleagues recently reported success with laminin-111 when they conducted Matrigel Sandwich type iPSC-CM differentiation experiments where the bottom substrate and overlay included different individual ECM proteins instead of only Matrigel [4]. They reported that iPSCs could differentiate into iPSC-CMs on a laminin-111 coating alone (achieving ≈ 80% cTnT+ cells after 15 days of differentiation). However, interestingly, an overlay of the cells with laminin-111 appeared to inhibit differentiation [4]. Others found that iPSC-CMs commercially purchased and then cultured on mouse laminin-111 could not support a confluent monolayer and produced more immature cardiomyocytes, determined by their displaying slower conduction velocities and lack of expression of a mature myofilament marker (cardiac troponin I, or cTnI), compared to Matrigel, though this may have been due to the iPSC-CMs having been differentiated on Matrigel and thus more adapted to this substrate [47].

While laminin-211 and -221 are both expressed in human cardiac tissue (Table 1) and serve as ECM ligands for integrin heterodimers also expressed in this tissue (α3β1 and α6β1), either limited studies have been performed testing these laminins individually for their ability to support CM differentiation, or it has been reported that they are unsupportive. Yap and colleagues [11] initially tested laminin-221 for its ability to support attachment of hESCs and found that this ECM protein alone could not support hESC attachment, though success was observed when combined with laminin-521, discussed below. Some studies have shown laminin-211 to not be expressed by hESC-CMs, potentially in alignment with the other functional studies here [66].

The most successful iPSC- or hESC-CM differentiation investigations conducted with laminins have been reported using laminins-511 and -521, with -521 alone or combined with -221 supporting the most efficient differentiations [5,6,11]. Laminin-521 and -511 have been shown to be suitable for culturing and maintaining pluripotent hESCs and iPSCs [6,53,67]. Burridge and colleagues tested several ECM proteins and peptides for their ability to support human iPSC-CM monolayers and iPSC-CM differentiation, including recombinant vitronectin, a synthetic vitronectin peptide, recombinant human laminin-521, and a truncated recombinant human laminin-511. Of the substrates tested, only the laminin-based matrices (laminin-521 and -511) supported long-term adhesion (>15 days) of motile iPSC-CM monolayers. At day 15 of differentiation, CM purities achieved on laminin-521, -511, and Matrigel were similar, all ≈85% TNNT2+ [6]. Later, Yap and colleagues similarly compared the ability of hESCs to differentiate into CMs on laminin-521 alone, laminin-521 combined with -211, or laminin-521 combined with -221 [11]. While laminin-521 alone and laminin-521 combined with -211 were both able to generate hESC-CMs (based on expression of TNNT2 and MYH1; ≈30–50% CMs), the purest hESC-CMs were generated on the combination of laminin-521 and -221 (≈80–90% CMs). The researchers further characterized hESC-CMs differentiated on laminin-521 and -221 in an entirely xeno-free protocol and found that, after a maturation process, the resultant hESC-CMs displayed relatively mature electrophysiological properties and appropriate drug responses (measured using single-cell patch clamp and multi-electrode array [MEA] analyses to measure drug compound responses), making them similar to those isolated from intact heart muscle, and overall demonstrating a coating of laminin-521 and -221 together to be promising to use for generating clinical-quality hESC-CMs [11]. However, this study did not include a Matrigel differentiation comparison, so it is difficult to say whether the laminin-521 and -221 protocol led to hESC-CMs with improved characteristics compared to standard Matrigel-based protocols. At the same time as Yap and colleagues, Sung and colleagues [5] identified laminin-521, performing individually, as one of the most suitable ECM proteins for hESC-CM differentiation, performing similarly successfully to CMs differentiated on Matrigel or Synthemax in terms of the survival of CMs over time, beating colony number, beating frequency (in its similarity to that of an adult heart), and purity of CMs produced, although cells differentiated on Synthemax had higher purity (Matrigel, Synthemax, and laminin-521 at 73%, 97%, and 60% CMs, respectively, after 14 days of differentiation, based on cTnT expression). In the Sung 2019 paper, laminin-521 in this way again outperformed laminin-511 (60% vs. 31% CMs, respectively), as well as collagen type I, fibronectin, and vitronectin [5]. Most recently, Zhang and colleagues reported successful iPSC-CM differentiation when iPSCs were differentiated on laminin-521, though, interestingly, when cells were overlayed with laminin-521 it appeared to inhibit differentiation [4].

Our group obtained similar iPSC-CM differentiation results when comparing a range of laminins, suggesting a combination of laminin-521 and -221 to be most supportive of early iPSC-CM differentiation, potentially out-performing Matrigel (Appendix A). We differentiated human iPSCs into iPSC-CMs on mouse laminin-111, human laminins -211, -221, -332, or -521 alone, human laminin-521 combined with -221, mouse collagen IV, or Matrigel (Materials and Methods provided in Appendix B). Laminins -332 and -521 alone or -521 combined with -221 best supported confluency during the first 11 days of differentiation, with collagen IV, laminin-211, and laminin-221 alone being least supportive (Appendix A). The mean day of contraction onset was earliest on the combination of laminin-521 with -221 (9.5 ± 0.5 days), followed by laminin-521 (11.0 ± 2.0 days) and then Matrigel (12.5 ± 3.5); interestingly, the lowest variability was observed on laminin-521 with -221, while Matrigel displayed the greatest variability (Appendix A). Contraction cell sheets were apparent for at least 32 days on these substrates (Appendix A) and cells expressed cardiac marker MLC-2a (Appendix A). This agrees with the other above findings, offering a combination of human laminin-521 with -221 as a substrate alternative to Matrigel that is not only a defined, purified, xeno-free substrate, but may even improve iPSC-CM differentiation results. 

### 3.3. Collagens

Collagens are common ECM proteins that play different roles during disease and development, providing physical stability for the cell, with different collagen types being expressed at different levels depending on tissue type, disease state, and stage of development [22,68]. In human cardiac tissue, the α3β1 integrin heterodimer is expressed, and while this integrin primarily interacts with laminins, it is also known to bind to and interacts with collagens, including collagen IV, which is additionally expressed in these tissues (Table 1) [54,60,61,62]. Both collagens IV and I interact with the integrin heterodimer α11β1, though the α11 appears to be lowly expressed in human cardiac tissue. Investigations have reported collagens I, IV, and XVIII to be expressed by some hESC-CMs [66], though ECM expression levels are likely impacted by the specific differentiation method employed.

While collagens used in complex, 3D scaffold or other synthetic fibrous matrix systems have been found supportive of CM differentiation and growth [69,70,71] and even maturation [72], and 2D, plate-based collagen coatings have been found supportive of iPSC differentiation into non-CM cell types [73,74,75,76], less has been reported on using 2D, plate-based collagen coatings to support CM differentiation specifically [4,5]. These limited, CM-focused, plate-based studies will be explored in the remainder of this section.

Collagen IV is of particular interest as it is the second most abundant component of Matrigel (following laminin-111), comprising approximately 30%. We previously found purified mouse collagen IV able to support expansion of undifferentiated human pluripotent stem cells, and did so better than human collagen IV, rat collagen I, or human collagen I (with the latter three performing similarly) [17]. However, Zhang and colleagues recently reported that iPSCs seeded on any one of several different substrates (Matrigel, laminin-111, -521, or fibronectin) when overlayed with collagen IV had inhibited abilities to differentiated into iPSC-CMs (≈0–15% CMs, based on percentage cTnT+ after 15 days of differentiation) [4]. In agreement with this, when our group used mouse collagen IV as a substrate for iPSC-CM differentiation, we found it performed relatively poorly; iPSC expansion during early iPSC-CM differentiation was relatively low compared to Matrigel and multiple laminins tested, and contractions were not observed (see Section 3.2. Laminins and Appendix A). Herron and colleagues similarly reported that iPSC-CMs commercially purchased and then cultured on mouse collagen IV could not support a completely confluent monolayer and produced relatively immature CMs, which (as they had also seen with mouse laminin-111) displayed slower conduction velocites and lacked expression of the mature myofilament marker cTnI, compared to CMs cultured on Matrigel, though this may have been due to the iPSC-CMs having been differentiated on Matrigel and thus better adapted to this substrate [47].

Collagen I has also been investigated for its ability to support hESC- and iPSC-CM differentiation [5], and was found unable to support early CM differentiation. All cells detached within 3–5 days of initiating differentiation, although other substrates tested in the same study (laminin-521, Matrigel, and Synthemax) could successfully support hESC-CM differentiation [5]. 

### 3.4. Fibronectin

Fibronectin is a glycoprotein in the myocardial ECM and it is primarily involved in cell adhesion. Fibronectin is expressed during mesoderm induction and fetal heart development, though cardiac expression decreases after birth [36,63]. Both fibronectin and vitronectin contain arginine-glycine-aspartate (RGD) binding motifs; this adhesive peptide is used in hydrogels and commercially available coated surfaces, such as Corning Synthemax products [50,77,78]. In human cardiac tissue, integrin heterodimers are expressed that bind fibronectin, including α3β1, α5β1, α8β1, and αVβ1 integrin heterodimers (Table 1). Other investigations have found fibronectin to be expressed by some hESC-CMs [66], though ECM expression levels are likely impacted by the specific differentiation method (including substrates) employed.

Interestingly, while fibronectin alone only poorly supports growth of pluripotent hPSCs, resulting in reduced confluencies over time compared to other ECM proteins [6,17], it can support hESC- and iPSC-CM differentiation. Sung and colleagues reported successfully creating hESC-CMs by differentiating hESCs on fibronectin [5]. Specifically, while vitronectin, collagen, and CellStart (a mixture of human albumin and fibronectin) coatings were unable to support hESC-CM differentiation, with all cells detaching within 3–5 days of initiating differentiation, hESC-CMs were produced on Matrigel, laminin-511, -521, Synthemax, vitronectin, or fibronectin, each individually. Differentiation of hESC-CMs on fibronectin yielded higher purities (36% CMs; based on percent of cells cTnT+ after 14 days of differentiation) than hESC-CMs derived on vitronectin or laminin-511 (22% or 31% CM, respectively), though lower purity compared to those derived on laminin-521, Matrigel, or Synthemax (60%, 73%, or 97% CMs, respectively) [5]. Most recently, Zhang et al., 2022 conducted Matrigel Sandwich-type iPSC-CM differentiation experiments with different substrates on the bottom or used as an overlay, and using this method identified a successful condition to be fibronectin on the bottom with fibronectin overlay (≈80% CMs based on cTnT expression at 15 days of differentiation) [4]. This purity of resultant CMs was the best of any conditions tested, with the other similarly performing conditions being Matrigel on the bottom with fibronectin overlay, and laminin-111 on the bottom with either no overlay, Matrigel overlay, or fibronectin overlay (all ≈80% CMs). Integrin inhibition studies revealed that the fibronectin-binding integrins α4β1 and αVβ1, but not the α5β1, may be important for supporting cardiac differentiation in these conditions. While others have similarly reported that hESC- or iPSC-CMs produced on an alternative substrate and then cultured on fibronectin could support a monolayer culture of hESC- or iPSC-CMs [47,58], maturation using fibronectin alone was unsuccessful (based on measuring conduction velocities) [47]. In Moyes et al., researchers found that the α5 and αV integrins in particular mediated hESC-CM adhesion and migration on fibronectin (likely through integrins α5β1 and αVβ1 specifically); combined with the findings reported in Zhang et al., 2022, this suggests integrin αVβ1 to play an important role in fibronectin-based hESC- and iPSC-CM culture systems. Overall, while mixed results have been reported using fibronectin to generate hESC- and iPSC-CMs, with relatively low purities observed and an inability to support maturation, the reported success in a Matrigel Sandwich-type approach by Zhang and colleagues suggests that fibronectin-based approaches warrant further investigation.

### 3.5. Vitronectin

Vitronectin is a glycoprotein expressed in the cardiac ECM (Table 1) involved in cell adhesion, expansion, and migration [79]. Like fibronectin, vitronectin contains an RGD motif through which it binds integrins; this peptide is used in hydrogels and synthetic surfaces such as Corning Synthemax products. Vitronectin interacts with many of the same integrin heterodimers as fibronectin; both fibronectin and vitronectin are bound by the integrin heterodimers α7β1, α8β1, and αVβ1 (Table 1). Vitronectin is additionally bound by αVβ5 [51,55].

While vitronectin interacts with many of the same integrin heterodimers as fibronectin, vitronectin is an established xeno-free, purified protein substrate that can maintain long-term undifferentiated hESC and iPSC culture, being a widely used feeder-free substrate option employed similarly to Matrigel [51,55]. Interestingly, while hESCs and iPSCs use β1 integrins for adhesion and proliferation on Matrigel, on vitronectin the αVβ5 integrin heterodimer is required for attachment, and both αVβ5 and β1 integrins are needed for proliferation [51].

Multiple groups have found vitronectin to be capable of supporting iPSC- or hESC-CM differentiation, though with inconsistent and mostly poor results, suggesting vitronectin to be incapable of supporting long-term CM attachment. Burridge et al. [6] found a recombinant human vitronectin could not support motile human iPSC-CM monolayers long-term; monolayers detached within ≈15 days of culture, indicating cell adhesion issues. Later, Sung and colleagues similarly reported that differentiation of hESCs into CMs on recombinant human vitronectin-coated dishes displayed poor cell attachment, with cells detaching within 3–5 days of initiating differentiation [5]. Despite these challenges, Burridge and colleagues were able to differentiate iPSC-CMs on recombinant human vitronectin with high purities (≈85% CMs; based on TNNT2+ expression, similar to the Matrigel control, after ≈15 days of differentiation). Sung and colleagues were also able to generate hESC-CMs on vitronectin, though this resulted in very low CM purities in their hands (22% CMs, based on cTnT expression after 14 days of differentiation), as well as low cell survival, yields, and beating frequency [5]. At the same time, Yap and colleagues investigated recombinant human vitronectin and similarly found while it could generate hESC-CMs, it did so poorly, resulting in the lowest purity of CMs generated, based on percentage of cells positive for markers TNNT2 and MYH1 (≈20–40% CMs; more supportive substrates being laminin-521, laminin-521 and -211 together, and laminin-521 and -221 together, as described above in Section 3.2, above) [11]. Burridge and colleagues speculated that lower expression levels of the αVβ5 integrin in human iPSC-CMs compared to undifferentiated human iPSCs may be responsible for the observed relatively decreased long-term adhesion of iPSC-CMs to vitronectin [6].

Despite these issues with vitronectin, multiple researchers have reported successfully differentiating hESC- or iPSC-CMs on the synthetic RGD vitronectin peptide with impressively high purities of CMs, producing results better than those seen with Matrigel and similar to, or better than, results produced by the other top-performing ECM-based substrate conditions of laminin-521 and -221 combined (Table 2). As seen with recombinant human vitronectin protein, Burridge et al. also found a synthetic vitronectin RGD peptide unable to support motile human iPSC-CM monolayers over time, observing detachment within ≈15 days of culture [6]. Despite these poor cell attachment issues, the researchers were still able to differentiate iPSCs into CMs on the synthetic vitronectin peptide, and the resultant iPSC-CMs resulted in CM purities even higher than that of CMs differentiated on the recombinant human vitronectin or Matrigel (≈90% for the peptide and ≈85% for recombinant vitronectin or Matrigel) [6]. They went on to passage and replate the iPSC-CMs on the vitronectin peptide and develop a xeno-free protocol in this way that produced relatively mature CMs, with 60% of cells expressing maturation marker MLC2V after 60 days. However, this appears to be an inefficient solution, and even in their study, laminins were found to be more supportive of iPSC-CMs and iPSC-CM differentiation but were not pursued due to cost (as discussed in Section 3.2, above) [6]. Sung and colleagues were similarly able to differentiate hESCs into high-purity, beating CMs on Synthemax II, with cultures possessing an impressive 97% CMs (based on positivity for cTnT expression after 14 days of differentiation); these were the purest CM cultures generated on any substrate condition tested in this study (which included collagen type I, fibronectin, and laminin-511) [5]. The hESC-CMs generated on Synthemax II were also the most successful generated in terms of hESC-CM survival, yields, and beating frequencies being similar to, or better than, those differentiated on Matrigel or laminin-521 [5].

## 4. Discussion

Significant progress has been made in identifying individual ECM proteins and synthetic peptide derivatives capable of supporting hESC- and iPSC-CM differentiation, which may lead the way to developing improved xeno-free differentiation protocols and scaffold systems, and a more biomimetic differentiation microenvironment. Such approaches would help overcome the limitations of using Matrigel, which is commonly employed in hESC- and iPSC-CM differentiation protocols but is a nonideal substrate for many applications as it is extracted from a tumor grown in mice (i.e., not xeno-free), contains a mixture of ECM proteins and growth factors that vary from lot to lot (i.e., not defined), and is not biomimetic, as discussed in Section 2.2. Protocol Limitations. Of all the ECM proteins, the extensive laminin family has been most heavily investigated for its ability to support hESC- and iPSC-CM differentiation, with only laminin-111, -511, and -521 alone found to be supportive, or laminin-521 in combination with -211 or -221 (Table 2). Limited successes have been reported with coatings of collagens, fibronectin, or vitronectin, though using a fibronectin coating with a fibronectin overlay led to promising results [4] that warrant additional attention to this ECM protein. Overall, the most promising of the ECM proteins or commercially available substrates reviewed here appear to be laminin-521 with -221 together or Synthemax (a synthetic vitronectin-based peptide coating) [5,6,11]. Both substrate systems resulted in highly pure (>80%) CM cultures, similar or better to efficiencies generated by differentiations on Matrigel (Table 2) [5,6,11]. This is important to note because one of the challenges in the hESC- and iPSC-CM differentiation field is generating highly pure populations of CMs. Furthermore, the resultant CMs from these substrate systems showed promise for overcoming another challenge in the field—immaturity of harvested hESC- and iPSC-CMs—as these CMs were demonstrated capable of becoming relatively mature. Because successes with maturing CMs on Matrigel have been reported (Table 2), performing differentiation on a combination of different ECM proteins together may potentially lead to successes in further improving maturation and, simultaneously, in reducing the currently lengthy differentiation culture times required.

A limitation of this review is the inherent challenge of comparing results between different studies, performed by different research groups, with varying base differentiation protocols. To help overcome this limitation with future investigations, the authors here recommend that researchers strive to always publish detailed methods protocols. Additionally, in comparative studies, considerable thought should be given to including and publishing data from all relevant control conditions; such control data (e.g., hESC- or iPSC-CM differentiation generated on Matrigel) serves as useful benchmark data for comparing findings between different investigations and potentially even normalizing data across different datasets. Because there are no standards for what timepoints during differentiation should be reported, or what CM markers are most representative, it would behoove researchers to report quantified data (i.e., flow cytometry analysis data or expression data quantified from immunocytochemistry assays) that are generated from multiple timepoints, using multiple CM markers, and tested with multiple cell lines. Additionally, because quantifying CM “maturation” is particularly challenging, it is recommended that researchers quantify data generated from multiple phenotypic and functional assays whenever reporting the maturity of CMs generated. Lastly, cell fold expansion values—or the number of CMs (or total cells) resulting from one initial iPSC—are rarely reported, but should be calculated and reported as a valuable quantitative metric for differentiation success; such values may also be useful for planning iPSC-based scale-up and clinical and commercial manufacturing needs. 

## 5. Conclusions

While laminin-521 and -221 together or Synthmax alone appear to be the most promising defined, xeno-free substrate coating systems supportive of hESC- or iPSC-CM differentiation, yielding highly pure CMs capable of maturation, side-by-side comparisons are needed to conclusively compare differentiation on these substrates to each other and Matrigel. Additional efforts are also needed to further improve these differentiation systems, including determining whether combinations of specific purified ECM proteins or derived peptides (potentially at varying concentrations or ratios) could reduce lengthy differentiation times and improve CM maturation. It is likely that no single ECM protein will be sufficient for use in optimal hESC- and iPSC-CM differentiation protocols, just as the in vivo microenvironment during cardiac development contains a diverse and changing subset of integrins and their complementary ECM binding partners. Furthermore, because the ECM microenvironment changes throughout CM development in vivo, the most supportive ECM microenvironment in vitro will likely similarly be required to be modulated throughout CM differentiation. If the current challenges of immaturity, lengthy culture times, and low purity could be overcome in improved hESC- and iPSC-CM differentiation protocols, it could enable more widespread use of these cells in cardiac regenerative medicine and disease modeling efforts.

## Figures and Tables

**Table 1 bioengineering-09-00720-t001:** Integrin and ECM protein expression in human cardiac tissue.

α Integrin Subunits Expressed *	β Integrin Subunits Expressed *	Primary ECM Integrin Ligand ^†^
α3	β1	**Laminin**-**211**, -213, -**221**, -311, -321, -323, -332, -**411**, -**421**, -**511**, -**521**, -523; **collagen IV**; fibronectin; thrombospondin-1
α5	β1	Fibronectin; fibrinogen; osteopontin
α6	β1	Laminin-111, -121, -**211**, -213, -**221**, -311, -321, -323, -332, -**411**, -**421**, -**511**, -**521**, -523
α7	β1	Laminin-111, -121, -**211**, -213, -**221**,
α8	β1	Fibronectin; **Vitronectin**
αV	β1	**Vitronectin**; fibronectin; fibrinogen; osteopontin
	β5	**Vitronectin**; osteopontin

* Integrin subunits reported to be expressed in human cardiac tissue [11,54,56,57,58]. Expression data on α1 and α4 integrin subunits was unavailable. ^†^ Binding interactions reported by [59,60,61,62]. Integrin ligands shown in bold were reported to be expressed in human cardiac tissue [54].

**Table 2 bioengineering-09-00720-t002:** Substrates investigated and compared for their ability to support hESC- or iPSC-CM differentiation.

Substrate	Purity of CMs Generated	CM Maturity Status	References
Laminin-111	≈80% without overlay; ≈0% with laminin-111 overlay; ≈80% with fibronectin overlay	Immature	[4,11]
Laminin-211	Unsupportive alone (requires laminin-521)	Not reported	[11]
Laminin-221	Unsupportive alone (requires laminin-521)	Not reported	[11]
Laminin-332	Not reported	Not reported	Appendix A
Laminin-511	31%; ≈85%	Not reported	[5,6]
Laminin-521	≈30–50%; ≈60%; ≈85%	Not reported	[4,6]
Laminin-521 + 211	≈30–50%	Not reported	[11]
Laminin-521 + 221	≈80–90%	Relatively mature following a maturation process	[11]; Appendix A
Collagen I	Unsupportive	Not reported	[5]
Collagen IV	≈0–15% with collagen IV overlay	Immature	[4,47]
Fibronectin	36%; ≈80% with fibronectin overlay	Immature	[4,5,47]
Vitronectin	22%; ≈20–40%; ≈85% with cell adhesion issues	Not reported	[5,6]
Synthemax (vitronectin peptide)	97%; ≈90% with cell adhesion issues	Relatively mature; 60% expressing MLC2V after 60 days	[5,6]
Matrigel	20% alone or 70% with Matrigel overlay; 73%; ≈85%; 80–98%	Mature following maturation period	[4,5,6,12,47]

## Data Availability

The data presented in this study are available in Appendix A.

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
