# Peer review of "Differentiating Human Pluripotent Stem Cells to Cardiomyocytes Using Purified Extracellular Matrix Proteins"

_bioengineering, 2022, doi:10.3390/bioengineering9120720_

Round 1

Reviewer 1 Report

GENERAL COMMENTS

The manuscript by Barnes A. et al. reviewed and discussed the defined, purified ECM proteins including laminins, collagens, fibronectin, vitronectin, Synthemax II and synthetic peptides that have been investigated in hPSC-cardiomyocytes differentiation in literatures. The review and discussion cover the specific ECM proteins and their integrin receptors, the hPSC-CMs differentiation protocols, the purity and further the maturity of the hPSC-CMs differentiated on different ECM surfaces. The authors also tested laminin-332 for hPSC-CMs differentiation which has not been reported in literature. The challenge in the hPSC-CM differentiation involves Matrigel as the ECM substrate which is not defined, a complex mixture of ECM proteins produced from Engelbreth-Holm-Swarm mouse sarcoma cells that exhibits batch-to-batch variability. The aim of the review is to promote using defined, purified, xeno-free ECM products for hPSC-CMs generation for clinical applications and regenerative medicine. The manuscript is a comprehensive review of the ECM proteins used in the hPSC-CMs differentiation, which play significant role yet not sufficiently investigated. The manuscript is well organized and clearly written, adequate references and informative.

SPECIFIC COMMENTS

1.      The discussion focused on the defined ECM proteins that express in human adult heart including laminins and collagens. However, the hPSC-CM differentiation is a multi-stage, dynamic process which is across multiple cell fate changes from pluripotent stem cells to mesodermal progenitors, cardiac progenitors, developing or immature cardiomyocytes and mature cardiomyocytes. So the ECM proteins during the differentiation is also dynamically changed, e.g. vitronectin or laminin-521 alone well support hPSC attachment and proliferation, but are not optimal for hPSC-CMs differentiation. On the other hand, laminin-221, laminin-211 and collagen IV are expressed relatively high in adult heart but these defined ECM proteins do not support hPSC attachment and expansion, therefore not suitable as the ECM substrate for hPSC-CM differentiation. For cardiac differentiation protocols a ECM substrate, or a combination of ECM proteins, or dynamically manipulation of ECM proteins, or applying different combination of ECM at different differentiation stages that allow attachment of the hPSCs and proliferation, and subsequent differentiation of mesodermal progenitors, cardiac progenitors and cardiomyocytes, and maturation of cardiomyocytes are likely needed. These should be discussed in the manuscript as well.

2.      The authors also compared the maturity of the hPSC-CMs generated from different groups that used different ECM substrates. However, these comparisons may not be appropriate due to different lengths of culture, media, functional measurement etc. Furthermore, the hPSC-CM populations differentiated using different ECM proteins as the substrates as well as from different protocols and different cell lines all contribute to different maturity of hPSC-CMs, so it may need more discussion.

3.      Line 155, manipulation of culture conditions to mature hPSC-CMs rather than lengthy culture have been reported. See Funakoshi et al. Generation of mature compact ventricular cardiomyocytes from human pluripotent stem cells. Nat Communications 2021.

Reviewer 2 Report

The review by Barnes et al. provides relevant insights on the use of several commercially available extracellular matrices in pluripotent stem cell (PSC) culture leading to cardiomyocyte differentiation. The impact of those ECMs in the purity and quality of the final cells is of great interest for disease modeling and regenerative medicine applications.

The review is well balanced and well structured, nonetheless this reviewer would like to point out areas of improvement:

General comments:

Cardiomyocyte maturity is mentioned often in the review, without specific metrics associated to it. Cardiomyocyte maturation involves metabolic changes, intracellular rearrangement and (electro)physiologic changes, between other aspects. So when comparing levels of maturity across experimental conditions or protocols it will be important to define which aspects and which baseline are being used for comparison in each instance.

The initial short introduction of the different ECMs starting on page 5 feels redundant and repetitive with their respective expanded sections that follow. This reviewer recommends consolidating all into one section per each ECM.

Specific comments:

Line 56: The authors mention that “Matrigel is extracted from the Engelbreth-Holm-Swarm mouse

Tumor”, while in fact it is harvested from a cell line derived from that tumor and cultured in industrial scale. That distinction feels important since as it reads it may give the impression that Matrigel is derived from a primary culture, which would be even more problematic from a reagent quality perspective and challenging to scale to industrial production.

Line 154: The authors mention that long culturing time is necessary to on the one hand achieve PSC-CM maturity, while on the other hand that can lead to genomic instability. It is important to distinguish that genomic instability may be more of a concern for PSC cultures, with high cell division rates, while differentiated PSC-CMs should be post-mitotic or minimally dividing and thus supposedly less prone to said instability. So if cardiomyocytes are generated from genomically stable PSCs, it may be unlikely that the long in vitro maturation process per se would drive genomic instability in the PSC-CMs. If the authors intend to suggest otherwise, that should be supported with appropriate relevant literature that differs from the citations provided.

Line 426: The authors refer to the potential role of a 3D collagen matrix in driving cardiomyocyte maturation, however the comparison made is between a 2D culture on Matrigel, vs the 3D constructs using collagen. The complexities of a 3D vs 2D environment may per se may drive differences on cardiomyocyte maturation, so it is difficult to isolate the role of collagen in this proposition. This section should be either expanded with more details that would hopefully provide more clarity, or removed.

Line 455: The authors mention that “hESC- or iPSC-CMs commercially purchased or produced on an alternative substrate and then cultured on fibronectin could support a monolayer culture

of iPSC-CMs.” Perhaps they meant that fibronectin can support monolayer formation of commercially available PSC-CMs produced in different substrates? That sentence should be re-written for clarity.

References: Several citations do not include their respective DOI. The bibliography style should be harmonized.

Reviewer 3 Report

Differentiating Human Pluripotent Stem Cells to Cardiomyocytes Using Purified Extracellular Matrix Proteins

Please confirm if the figures within the review are original and did not publish elsewhere.

More summary tables are recommended to be added to summarize the previous studies related to the review topic

Reviewer 4 Report

This review deals with an important topic for cardiac regenerative medicine. But, it needs more organization and editing to be more informative and deepen the knowledge on the top as follows:

1.      Editing of the language of the manuscript by a native English speaker is highly needed, with a special focus on the following points:

·         The writing style should be formal from the third-person perspective. Do not use we (E.g. line 17 )

·         It is not preferable to begin sentences with abbreviations like hESC- in line 306.

2.      The section 2.1. Overview of protocols need to be more expanded to give satisfactory background on the protocols with schematic figures of them to be the background of the second section of limitations.

3.      Throughout the review, the authors mentioned many times “many researchers” and then the paragraph ends with a reference to one study, like line 23. Please, revise the reference's citation.

4.      Lines 259-264: add the reference for this information.

5.      I would recommend the authors give more discussions on lessons learned from the state of the science and challenges in this field in a new separate discussion section to show the manuscript's contribution more clearly.

6.      In the figure legends: the full term of all abbreviations used should be clarified.

7.      The tables and figures are not satisfactory.

Round 2

Reviewer 4 Report

No further comments are to be addressed